# Risk of Lymphedema and Death after Lymph Node Dissection with Neoadjuvant and Adjuvant Treatments in Patients with Breast Cancer: An Eight-Year Nationwide Cohort Study

**DOI:** 10.3390/healthcare11131833

**Published:** 2023-06-23

**Authors:** Ye-Seul Lee, Yu-Cheol Lim, Jiyoon Yeo, Song-Yi Kim, Yoon Jae Lee, In-Hyuk Ha

**Affiliations:** 1Jaseng Spine and Joint Research Institute, Jaseng Medical Foundation, 540 Gangnam-daero, Seoul 06110, Republic of Korea; yeseul.j.lee@gmail.com (Y.-S.L.); hmh6692@gmail.com (Y.-C.L.); jyyeo0605@gmail.com (Y.J.L.); 2Department of Economics, Korea University, Seoul 02841, Republic of Korea; goodsmile8119@gmail.com; 3Department of Acupoint and Anatomy, College of Korean Medicine, Gachon University, Seongnam 13120, Republic of Korea; songyi@gachon.ac.kr

**Keywords:** breast cancer, lymphedema, sentinel lymph node biopsy, axillary lymph node dissection, chemotherapy, radiotherapy, longitudinal observational cohort study

## Abstract

Knowledge on the impact of neoadjuvant and adjuvant treatments on post-surgery lymphedema (LE) in patients with breast cancer is limited due to methodological limitations and an insufficient sample size. We investigated the risk of LE in patients going through long-term anticancer treatment regimens using a national cohort from the Korean National Health Insurance Service database from 2011–2013. Incidence rate ratio, Kaplan–Meier analysis, and Cox proportional regression analysis were performed. A total of 39,791 patients were included. While minimal lymph node dissection (SLNB) reduced the risk of LE (hazard ratio [HR] 0.51) as expected, neoadjuvant chemotherapy (NAC) followed by SLNB did not reduce the risk. Adjusting for adjuvant chemotherapy (AC) as time-varying exposure decreased the risk of LE in the SLNB group (HR 0.51), but not the mortality risk (HR 0.861). A longer duration of NAC, especially taxane-based, combined with SLNB reversed the effect and increased risk of LE. The findings highlight the importance of not only early surveillance before and after surgery, but also long-term surveillance during adjuvant treatment by surgeons and oncologists in order to reduce the risk of LE.

## 1. Introduction

Breast cancer is the most common cancer worldwide, with 2,261,419 new cases reported in 2020 [1,2]. It is also the fifth cause of cancer death following lung, colorectal, liver, and stomach cancer [1]. The 5-year survival rate of women with breast cancer is as high as 90.8% (2013–2019) in the US [3] and 93.0% (2014–2018) in the Republic of Korea [2], owing to recent advances in screening methods, early diagnosis, and multimodal therapies [4,5]. However, treatment complications remain a substantial threat to the quality of life of patients with breast cancer [6].

Lymphedema (LE) is one of the most common complications associated with breast cancer treatment [7,8]. The prevalence of LE and its association with treatment modalities show a variance of 7% to >55% [9,10,11], which could possibly be attributed to the methodological differences in the definitions of these treatments and the differences in individual patient characteristics, such as body mass index (BMI) and older age [7,12]. LE might present in the postoperative stage or 2–3 years after axillary dissection [9,13]. Although complete decongestive therapy (CDT) or manual drainage is proven to be effective [14,15,16,17], definite treatment does not exist.

Available evidence on the association between treatment modalities for breast cancer and LE risk is inconclusive due to discrepancies. A number of studies demonstrated a lower LE risk from minimal axillary dissections [11,18,19]; meanwhile, others reported contrary results [20,21,22]. Similarly, studies on the association between the LE risk and neoadjuvant chemotherapy (NAC), adjuvant chemotherapy (AC), or radiotherapy have yielded mixed results [23,24,25,26]. This divergence may be attributed to the difficulties in investigating multiple treatment regimens in a study population with a limited sample size.

Hence, this retrospective, nationwide, 8-year cohort study aimed to investigate the association between the LE risk from chemotherapy and radiotherapy before and after axillary dissection, in neoadjuvant and adjuvant settings, using multivariate survival analyses including time-varying exposures. Our research question was the influence of long-term anticancer therapy in addition to the effect of minimal axillary dissection on the risk of LE, and time-varying exposures were used in the analyses.

## 2. Materials and Methods

### 2.1. Data Source

This retrospective, nationwide, cohort study used data from the national health claims database established by the National Health Insurance Service (NHIS) of Korea. This database includes data from the population subscribed [27] to the NHIS (97%), a single insurer providing universal health coverage, and from those receiving medical aid (3%), from 2002–2019 [28]. This database is considered nationally representative and includes the date of death of the population from Korean Statistics. The related materials and metadata are publicly available on the National Health Insurance Data Sharing Service homepage (http://nhiss.nhis.or.kr (accessed on 30 March 2023)).

### 2.2. Cohort Design and Study Population

The study was designed based on a previous clinical trial [29] and compared the outcomes of lymph node dissection of varying invasiveness, chemotherapy, and radiotherapy in neoadjuvant and adjuvant settings. The study included patients with breast cancer who were diagnosed based on the International Classification of Diseases, 10th revision code C50 (Figure 1), from 1 January 2011 to 31 December 2013. The time period was selected to ensure a wash-out period and a minimum of 4 years of follow-up [9]. Rigorous eligibility and exclusion criteria were applied to ensure patient homogeneity. Exclusion criteria included patients with (1) a history of breast cancer since 2002; (2) a history of any cancer throughout the year preceding the entry date; (3) a history of mastectomy, lumpectomy, or lymphadenectomy prior to diagnosis; (4) missing surgical records for 2 years following diagnosis; (5) those who underwent lymphadenectomy but not mastectomy and lumpectomy; (6) those who were diagnosed with lymphedema prior to cancer diagnosis or surgery, or on the day of surgery; (7) CDT prior to diagnosis or surgery, or on the day of surgery, and (8) those who were diagnosed with ductal carcinoma in situ (DCIS). The index date of the cohort was identified as the day of surgery. Each patient was followed for 4 years from the index date, or until they developed lymphedema or died, depending on the model.

An intention-to-treat design was applied for the type of surgical dissection. The intervention was defined as composites of mastectomy/lumpectomy along with axillary dissections. Patients who underwent either sentinel lymph node biopsy (SLNB) and axillary lymph node dissection (ALND) were categorized into respective groups based on their surgical information found in the NHIS records. Patients with multiple lymph node dissections or breast dissection within 60 days of the diagnosis were grouped into the reoperation group (Appendix A). Patients who only underwent lymphadenectomy due to potential occult or recurrent breast cancer diagnosis were excluded from the study.

### 2.3. Outcome and Covariates

The primary outcome of the study was LE incidence within 4 years of surgery. The secondary outcome was all-cause mortality during the follow-up period. NAC, neoadjuvant radiotherapy (NAR), AC, and adjuvant radiotherapy (AR) were included as covariates in the analysis. The baseline characteristics of the patients included information on the demographic variables such as sex, age, type of health insurance, the Charlson comorbidity index (CCI), and 18 prespecified comorbidities that occurred within the 1-year period preceding breast cancer diagnosis (Appendix A).

Radiotherapy and chemotherapy regimens were identified by their intervention codes, found in the NHIS records of the hospitals. The duration of AR or AC regimens was calculated from the day of surgical dissection until the incidence of the outcome for patients who developed LE or died during the follow-up period, and until the end of the follow-up period for patients who experienced neither the primary nor secondary outcomes.

### 2.4. Statistical Analyses

A descriptive analysis was conducted to identify the cohort’s baseline characteristics. The propensity score (PS) was calculated as the probability of receiving SLNB or ALND along with breast dissection. The PS calculation was performed using demographic factors, duration of neoadjuvant treatments, the CCI, and prespecified comorbidities as classification variables, and the duration of neoadjuvant treatments as an exact condition. PS matching was performed using the greedy nearest neighbor method with one matched control for each patient with a caliper of 0.01.

The incidence rate ratio (IRR) was computed, and Cox proportional hazard models were used to calculate the LE risk. Four multivariate Cox regression models were generated to adjust for demographic factors, the CCI, and NAC, in addition to the type of surgery performed. A time-varying Cox regression model was generated to adjust for adjuvant cancer therapies during the follow-up period as time-varying exposures to avoid violation of the proportional hazards assumption [30]. Kaplan–Meier curves were generated to estimate the time duration from the cohort index date to the date of the LE event. A two-sided *p*-value of <0.05 was considered significant. Statistical analyses were conducted using the SAS version 9.4 (SAS Institute, Cary, NC, USA).

### 2.5. Sensitivity Analysis

We conducted seven prespecified sensitivity analyses. The details are described in Appendix A.

## 3. Results

### 3.1. Patient Characteristics

Among the 56,618 patients newly diagnosed with breast cancer in Korea between 2011 and 2013, 39,791 satisfied the inclusion criteria (Figure 2). Most patients included in the study were between 45 and 54 years of age (Table 1) (mean age at the baseline, 51.57 [SD = 10.99] years). Almost all the patients (98%) were enrolled in the NHIS and their CCI was 0. The patients in the ALND group were likely to have received NAC (SLNB group: 5.8%; ALND group: 21.7%; reoperation group: 8.1%), with most patients receiving one to four cycles (ALND group: 13.3%). In the ALND group, 10.0% of the patients received a taxane-based NAC, compared to 1.5% and 6.3% in the SLNB and reoperation groups, respectively.

Among the participants, 65.4% (*n* = 26,036) received AR until LE incidence or during the follow-up period (Table 1); this percentage increased to 71.0% (*n* = 28,248) when patients were followed-up until death or the end of the follow-up period (Appendix A). Meanwhile, 62.5% of the patients (*n* = 24,854) received AC until LE incidence or during the follow-up period (Table 1); this increased to 64.1% (*n* = 25,495) when followed up until death or the end of the follow-up period (Appendix A). Of all patients, 23.8% received a taxane-based AC.

After PS matching, 6688 patients were included in the SLNB and ALND groups (Table 1). Notably, none of these patients received NAR. Patients in the SLNB group showed a slightly higher incidence of comorbidities, including osteoporosis and sleep disorders.

### 3.2. Risk of Developing LE by Type of Surgery and Adjuvant Treatment

In the matched cohort, the SLNB group had a lower risk of developing LE (IRR 0.496, 95% confidence interval (CI) 0.450–0.545, Appendix A; unadjusted hazard ratio (HR) 0.508, 95% CI 0.462–0.559, Table 2; Figure 3A).

The effect of SLNB in lowering the LE risk was observed in all multivariate Cox proportional hazards models (Table 2). However, NAC (HR 1.866, 95% CI 1.597–2.182, Figure 3C), especially taxane-based NAC (HR 2.527, 95% CI 1.944–3.286, Appendix A), increased the LE risk, but that in combination with SLNB reversed the risk (Table 2).

Time-dependent analysis of AR showed a significantly lowered LE risk in the SLNB group (HR 0.513, 95% CI 0.466–0.565; Table 3) and the time-dependent exposure of AR was not associated with the LE risk (β = 0.002, *p* = 0.265; Table 3).

Time-dependent analysis of AC showed a significantly lowered LE risk in the SLNB group (HR = 0.514, 95% CI 0.467–0.566), with an increased risk with the increasing number of AC cycles (β = 0.059, *p* < 0.001; Table 3). Notably, the combination of SLNB with >5 cycles of AC reversed the risk.

### 3.3. All-Cause Mortality by the Type of Surgery

The patients in the SLNB group had a lowered risk of death (IRR 0.793, 95% CI 0.637–0.986, Appendix A; unadjusted HR 0.793, 95% CI 0.637–0.986, Table 2; Figure 3B) and the significance was retained even after adjusting for other covariates, as seen in all multivariate Cox proportional hazards models (Table 2). Contrastingly, the risk of death increased in patients aged ≥65 years (Model 2, Table 2), those who availed medical aid (HR 2.206, 95% CI 1.491–3.264, Model 2, Table 2), and those with a CCI of 3 (HR 1.612, 95% CI 1.082–2.403, Model 3, Table 2).

NAC (HR 2.521, 95% CI 1.761–3.608, Table 2, Figure 3D), especially taxane-based NAC (HR 6.093, 95% CI 3.768–9.853, Appendix A), was associated with an increased risk of death, which was also observed in patients who underwent 5–14 sessions of NAC (HR 6.792, 95% CI 4.140–11.144; Table 2). Notably, the decreased risk was reversed in patients who had NAC in addition to SLNB.

In the time-varying regression analysis, death is no longer associated with the type of surgery. AR was not associated with the risk of death (HR 0.994, 95% CI 0.987–1.001); however, AC decreased the risk of death (HR 0.955, 95% CI 0.942–0.968). Further, increasing the AR duration did not affect the all-cause mortality, whereas the increase in the number of AC cycles significantly decreased this risk (β = −0.046, *p* < 0.001; Table 3).

### 3.4. Sensitivity Analysis

Incorporating the prespecified comorbidities in the multivariate Cox regression model yielded similar results as those from before matching (Appendix A). Cardiovascular diseases (CVD, HR 1.491, 95% CI 1.164–1.529) and osteoarthritis (HR 1.199, 95% CI 1.050–1.370) were associated with a higher LE risk. Chronic liver diseases (CLD, HR 1.718, 95% CI 1.047–2.819), dementia, Alzheimer’s disease (HR 2.825, 95% CI 1.458–5.475), and renal failure (RF, HR 5.342, 95% CI 2.973–9.601) were associated with a higher risk of death (Appendix A).

The study findings derived from the cohort prior to matching were consistent with the main findings. NAR was not associated with the LE risk (HR 1.084, 95% CI 0.706–1.665) but was associated with a higher risk of death (HR 8.432, 95% CI 5.768–12.327, Appendix A).

Comparison between groups showed a lowered LE risk in the SLNB group (unadjusted HR 0.312, 95% CI 0.260–0.374). However, the all-cause mortality was similar between the groups (Appendix A).

Findings derived from regrouping the cohort based on the type of surgical resection received within 60 days were consistent with the main findings on the LE risk (unadjusted HR 0.381, 95% CI 0.352–0.413, Appendix A). The risk of all-cause mortality was substantially lower in patients in the SLNB group (unadjusted HR 0.595, 95% CI 0.502–0.704, Appendix A).

Finally, the type of surgery and adjuvant therapy showed no effect on the risks of LE and death (Appendix A). Univariate analysis of the subgroup of patients with events showed that the incidence of LE and death escalated rapidly in two stages in response to AR duration but increased gradually in response to AC duration (Appendix A).

## 4. Discussion

This study evaluated the impact of long-term treatment regimens, including surgery, radiotherapy, and chemotherapy in neoadjuvant and adjuvant settings, on the LE risk from all causes using a national cohort. Both neoadjuvant and adjuvant chemotherapies, especially taxane-based NAC, increased the LE risk in a dose-dependent manner, such that the combination of NAC and SLNB led to a higher risk of LE than ALND alone. In addition, underlying comorbidities such as CVD, osteoarthritis, and a CCI of ≥3 moderately increased the LE risk.

Time-varying survival analysis was performed to account for the subsequent influence of adjuvant cancer therapies on the LE risk without violating the proportional hazards assumption, which revealed that LE risk increased with an increase in the number of AC cycles. This is consistent with the findings from previous clinical trials and retrospective studies [13,19,31,32]. Meanwhile, this study demonstrated that AR did not significantly influence the LE risk, similar to the findings from previous studies [7].

Treatment strategies for breast cancer [33,34,35,36] suggest axilla staging during initial surgical resection, which was the standard treatment procedure for almost three decades [35]. During the last decade, the surgical trend shifted from ALND to SLNB to decrease the risk of LE based on a number of studies [11,18,19], although few studies have raised concerns about this shift [20,21]. In this study, SLNB reduced the LE risk, as shown by multivariate regression analysis in models adjusted for both baseline characteristics and neoadjuvant treatments and in time-varying regression models with adjuvant treatments. This decrease in the LE risk remained significant even after redefining the intervention into a composite of surgical dissections within 60 days in comparison with the reoperation group.

The findings also showed that a treatment regimen of SLNB combined with an increased duration of taxane-based NAC might not reduce the LE risk, compared to that of ALND and/or mastectomy without NAC. NAC is usually administered to reduce the tumor size to make it operable or to enable less invasive operations [37]; however, it may reduce the advantage of SLNB in minimizing the adverse effects. The present study’s findings call for caution during symptom management of patients who received breast conserving surgery or SLNB after a long duration of NAC, especially if taxane-based. Although limited studies are available on the association of NAC and LE due to difficulties in assessing such modalities retrospectively, recent studies support our findings that NAC [7,38], especially if taxane-based and administered for a long duration [7,39], increased the LE risk.

It is necessary to consider the difference in disease severity between patients who underwent SLNB and ALND when evaluating the LE risk. Even after excluding patients with DCIS, SLNB is recommended in patients with T1/T2, multicentric tumors, and certain T3, e.g., large tumors, but against T4 and inflammatory tumors [40]. Additionally, those who received SLNB followed by ALND within 60 days were included in this study. Overall, patients who underwent ALND are more likely to have poorer prognoses. In this study, the mortality risk of the patients in the SLNB group was not lower than those in the ALND group after considerable cycles of NAC and/or AC were administered.

Notably, the differences in the mortality risk between the two groups diminished when the time-varying regression model was implemented, indicating the efficacy of appropriate adjuvant treatments in decreasing the mortality risk. In addition, the effect of the type of lymph node dissection remained significant in the time-varying regression model. Thus, the type of lymph node dissection influences the risk of LE but has no significant impact on long-term survival if patients receive adjuvant treatments following surgery. Time-varying exposures to AC, on the contrary, increased the LE risk while reducing the risk of mortality, highlighting the need for diligent monitoring of patients receiving AC to avoid adverse events and maximize their chances of survival.

Finally, studies on the risk factors of LE report that BMI [12,19,41,42] and age [7,25,41] might be associated with LE, underscoring the multifactorial nature of LE despite their inconsistent findings. In contrast, our study did not find an association between age and LE, which is in line with other studies stating that age cannot be a standalone surrogate for lymphedema risk and should be complemented by other metrics [7,41]. However, CVD and osteoarthritis prior to breast cancer diagnosis [43] were associated with a higher LE risk [7]. As patients with breast cancer in Korea are relatively younger [44], with lower BMIs [42,45], these findings might provide some basis for an early assessment. Further studies are crucial to understand the mechanisms of such associations and ensure their generalizability. This study also showed that dementia, Alzheimer’s disease, CLD, and RF were associated with an increased risk of death, in corroboration with findings of previous studies on the US population [46].

This study has some limitations. First, identifying patients with a breast cancer history before 2002 (the conception of the database) was not possible. Second, pharmacological profiles were not evaluated; thus, we could not identify the prescriptions of patients with CVD and osteoarthritis and could not distinguish whether the impact of such comorbidities could be attributed to the use of medications or innate pathological mechanisms. Third, specific treatment regimens and the types of chemotherapy medications administered were not categorized. Finally, only one definition of LE was adopted. While we considered adopting a composite definition of LE diagnosis and CDT treatments, it was not included as the prescriptions for CDT were often at the discretion of physicians rather than based on strict diagnostic standards. Therefore, identifying patients with severe LE using CDT was not possible. Lastly, the inherent bias of a retrospective cohort analysis remain as this study is not based on prospective clinical trials. The major strengths of this study include the use of a national cohort to minimize the potential selection bias and using a time-varying regression model, which allowed us to adjust for adjuvant treatments without violating the proportional hazards assumption. This study design allowed for the investigation of the effect of long-term anticancer treatments on the risk of LE, a long-awaited research question that could not have been addressed using conventional clinical trial settings.

## 5. Conclusions

In this retrospective cohort study, LE risk decreased in patients with breast cancer who underwent SLNB; however, this effect diminished in patients who received >4 cycles of NAC before SLNB. While ALND is usually performed in patients who are in advanced stages or with inflammatory tumors, no significant difference in mortality was observed compared to SLNB when time-varying exposures of adjuvant treatments were incorporated into the model. On the contrary, the reduced LE risk in the patients who underwent SLNB was significant even after adjusting for time-varying exposures; however, an increase in the number of AC cycles significantly increased the risk. The study findings emphasize not only the need for early surveillance before and after surgery but also long-term surveillance during adjuvant treatment by surgeons and oncologists to minimize the risk of LE.

## Figures and Tables

**Figure 1 healthcare-11-01833-f001:**
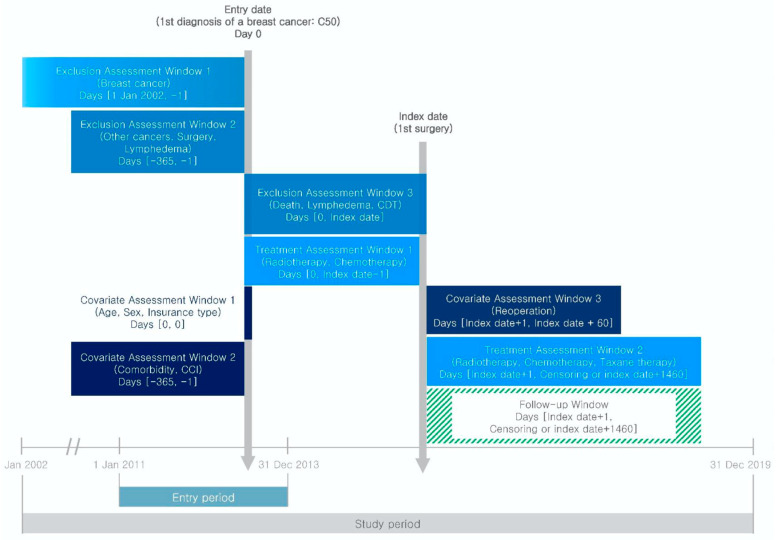
Study design. Abbreviations: CDT, complete decongestive therapy; CCI, Charlson comorbidity index.

**Figure 2 healthcare-11-01833-f002:**
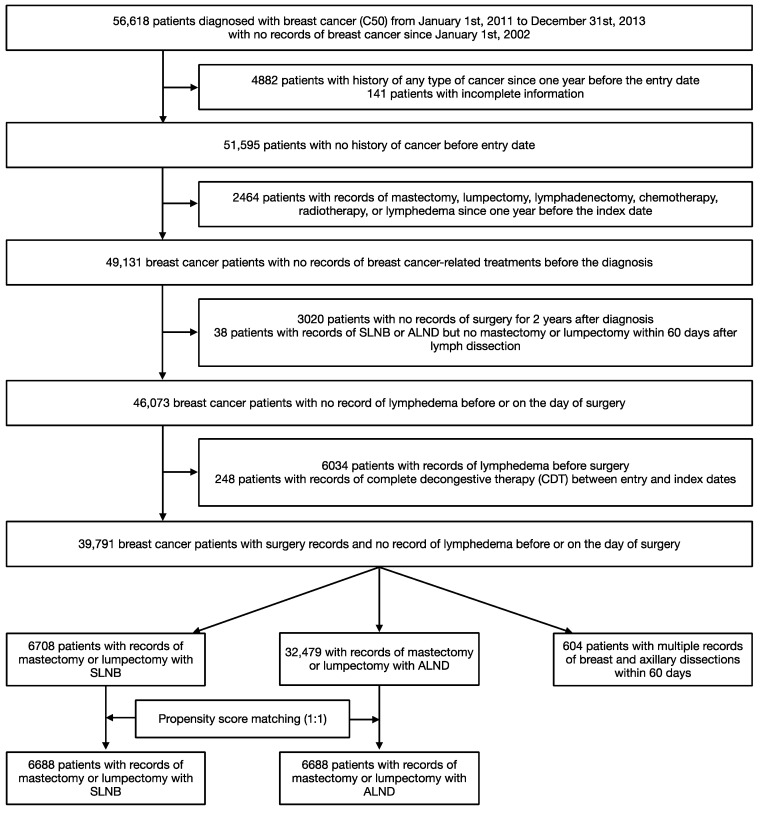
Study population. Abbreviations: ALND, axillary lymph node dissection; SLNB, sentinel lymph node biopsy.

**Figure 3 healthcare-11-01833-f003:**
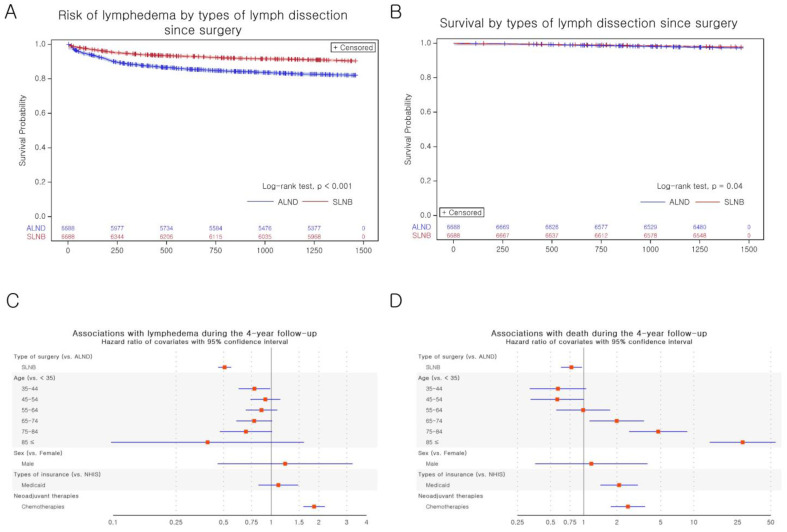
Kaplan–Meier survival curves and forest plots of PS matched cohorts for the risk of LE and all-cause mortality. (**A**)Kaplan–Meier survival curves for the risk of LE; (**B**) Kaplan–Meier survival curves for all-cause mortality; (**C**) forest plot of multivariate Cox regression adjusted for baseline characteristics and NAC for the risk of LE; (**D**) forest plot of multivariate Cox regression adjusted for baseline characteristics and NAC for all-cause mortality. Abbreviations: PS, propensity score; LE, lymphedema; ALND, axillary lymph node dissection; SLNB, sentinel lymph node biopsy; NHIS, National Health Insurance Service; NAC, neoadjuvant chemotherapy.

**Table 1 healthcare-11-01833-t001:** Baseline characteristics of the study population.

Variables	Before Propensity Score Matching							After Propensity Score Matching				
	SLNB (*n* = 6708)		ALND (*n* = 32,479)		Reoperation within 60 Days ^1^ (*n* = 604)		*p*-Value	SLNB (*n* = 6688)		ALND (*n* = 6688)		*p*-Value
	N	%	N	%	N	%		N	%	N	%	
Age group, years							<0.0001					0.956
<35	274	4.1	1355	4.2	38	6.3		270	4.0	282	4.2	
35–44	1498	22.3	7405	22.8	161	26.7		1494	22.3	1495	22.4	
45–54	2701	40.3	12,514	38.5	240	39.7		2693	40.3	2677	40.0	
55–64	1404	20.9	6804	21.0	113	18.7		1401	21.0	1391	20.8	
65–74	627	9.4	3326	10.2	43	7.1		627	9.4	645	9.6	
75–84	182	2.7	990	3.1	9	1.5		181	2.7	182	2.7	
≥85	22	0.3	85	0.3	-	-		22	0.3	16	0.2	
Sex							0.677					0.369
Female	6690	99.7	32,370	99.7	602	99.7		6670	99.7	6675	99.8	
Insurance type							0.020					0.375
NHI	6532	97.4	31,464	96.9	593	98.2		6513	97.4	6529	97.6	
Medical Aid	176	2.6	1015	3.1	11	1.8		175	2.6	159	2.4	
CCI							0.336					0.467
0	5655	84.3	27,543	84.8	522	86.4		5641	84.4	5688	85.1	
1–2	828	12.3	3834	11.8	69	11.4		822	12.3	776	11.6	
3–	225	3.4	1102	3.4	13	2.2		225	3.4	224	3.4	
Neoadjuvant radiotherapy							0.078	-	-	-	-	-
Yes	6	0.1	66	0.2	-	-						
Number of sessions							0.525					
1–20	4	0.1	43	0.1	-	-						
21–40	2	0.0	22	0.1	-	-						
41–	-	-	1	0.0	-	-						
Neoadjuvant chemotherapy							<0.0001	379	5.7	379	5.7	1.000
Yes	386	5.8	7032	21.7	49	8.1						
Number of sessions							<0.0001					1.000
1–4	294	4.4	4326	13.3	13	2.2		292	4.4	292	4.4	
5–14	90	1.3	2647	8.2	36	6.0		87	1.3	87	1.3	
15–	2	0.0	59	0.2	-	-		-	-	-	-	
Neoadjuvant taxane treatment							<0.0001					0.829
Yes	100	1.5	3234	10.0	38	6.3		96	1.4	99	1.5	
Comorbidities												
Anemia	169	2.5	650	2.0	13	2.2	0.026	169	2.5	136	2.0	0.056
Anxiety disorder	161	2.4	660	2.0	18	3.0	0.052	158	2.4	127	1.9	0.063
Cardiovascular disease	199	3.0	963	3.0	23	3.8	0.481	199	3.0	182	2.7	0.377
Cerebrovascular disease	131	2.0	647	2.0	10	1.7	0.828	131	2.0	133	2.0	0.901
Chronic back pain	1839	27.4	8966	27.6	136	22.5	0.021	1833	27.4	1822	27.2	0.831
Chronic liver disease	217	3.2	954	2.9	17	2.8	0.414	214	3.2	207	3.1	0.729
Chronic obstructive pulmonary disease	27	0.4	134	0.4	1	0.2	0.639	26	0.4	31	0.5	0.507
Dementia & Alzheimer	22	0.3	120	0.4	-	-	0.291	22	0.3	18	0.3	0.527
Depressive disorder	158	2.4	696	2.1	18	3.0	0.229	156	2.3	127	1.9	0.081
Diabetes	428	6.4	2104	6.5	27	4.5	0.135	427	6.4	402	6.0	0.370
Hyperlipidemia	361	5.4	1767	5.4	42	7.0	0.257	359	5.4	364	5.4	0.848
Hypertension	1124	16.8	5873	18.1	104	17.2	0.033	1121	16.8	1153	17.2	0.461
Osteoarthritis	953	14.2	4688	14.4	76	12.6	0.402	950	14.2	943	14.1	0.862
Osteoporosis	276	4.1	1313	4.0	26	4.3	0.919	276	4.1	228	3.4	0.029
Renal failure	39	0.6	127	0.4	4	0.7	0.063	39	0.6	34	0.5	0.557
Rheumatoid arthritis	94	1.4	439	1.4	12	2.0	0.400	92	1.4	85	1.3	0.596
Schizophrenia	21	0.3	101	0.3	1	0.2	0.814	21	0.3	16	0.2	0.410
Sleep disorder	176	2.6	727	2.2	11	1.8	0.117	172	2.6	135	2.0	0.033
Adjuvant radiotherapy ^1^							<0.0001					0.834
Yes	4781	71.3	20,878	64.3	377	62.4		4766	71.3	4755	71.1	
Number of sessions							<0.0001					0.988
1–20	1563	23.3	4790	14.8	80	13.3		1552	23.2	1537	23.0	
21–40	3194	47.6	15,884	48.9	291	48.2		3190	47.7	3195	47.8	
41–	24	0.4	204	0.6	6	1.0		24	0.4	23	0.3	
Adjuvant chemotherapy ^1^							<0.0001					0.704
Yes	3284	49.0	21,238	65.4	332	55.0		3279	49.0	3257	48.7	
Number of sessions							<0.0001					0.943
1–4	1368	20.4	5849	18.0	88	14.6		1367	20.4	1366	20.4	
5–14	1266	18.9	10,437	32.1	188	31.1		1264	18.9	1263	18.9	
15–	650	9.7	4952	15.3	56	9.3		648	9.7	628	9.4	
Adjuvant taxane treatment ^1^							<0.0001					<0.0001
Yes	457	6.8	8851	27.3	169	28.0		455	6.8	1104	16.5	
Number of sessions (mean ± SD)	0.36 ± 1.68		1.33 ± 2.76		1.26 ± 2.42			0.36 ± 1.68		0.80 ± 2.17		

^1^ Followed up until the development of LE or the end of the 4-year follow-up. Abbreviations: SLNB, sentinel lymph node biopsy; ALND, axillary lymph node dissection; NHI, National Health Insurance; CCI, Charlson comorbidity index; SD, standard deviation.

**Table 2 healthcare-11-01833-t002:** Univariate and multivariate analyses of the association of baseline characteristics with lymphedema and death.

	Associations with Lymphedema during the Four-Year Follow-Up					Associations with Death during the Four-Year Follow-Up				
Covariates	Model 1 (Unadjusted)	Model 2 ^1^	Model 3 ^2^	Model 4 ^3^	Model 5 ^4^	Model 1 (Unadjusted)	Model 2 ^1^	Model 3 ^2^	Model 4 ^3^	Model 5 ^4^
	HR (95% CI)	HR (95% CI)	HR (95% CI)	HR (95% CI)	HR (95% CI)	HR (95% CI)	HR (95% CI)	HR (95% CI)	HR (95% CI)	HR (95% CI)
Type of surgery (vs. ALND)										
SLNB	0.508 (0.462–0.559) ***	0.507 (0.461–0.559) ***	0.507 (0.460–0.558) ***	0.507 (0.461–0.558) ***	0.507 (0.460–0.558) ***	0.793 (0.637–0.986) *	0.777 (0.625–0.968) *	0.776 (0.623–0.966) *	0.773 (0.621–0.963) *	0.777 (0.624–0.968) *
Age, years (vs. <35)										
35–44		0.739 (0.588–0.929) **	0.740 (0.589–0.931) *	0.783 (0.622–0.985) *	0.796 (0.632–1.002)		0.525 (0.291–0.944) *	0.525 (0.292–0.946) *	0.583 (0.323–1.053)	0.640 (0.353–1.160)
45–54		0.863 (0.695–1.072)	0.859 (0.691–1.068)	0.918 (0.738–1.142)	0.939 (0.754–1.169)		0.515 (0.296–0.896) *	0.511 (0.294–0.890) *	0.575 (0.329–1.002)	0.658 (0.374–1.157)
55–64		0.815 (0.648–1.024)	0.797 (0.634–1.002)	0.868 (0.690–1.092)	0.889 (0.706–1.120)		0.879 (0.502–1.538)	0.856 (0.488–1.500)	0.988 (0.563–1.735)	1.145 (0.646–2.031)
65–74		0.716 (0.553–0.928) *	0.687 (0.529–0.892) **	0.780 (0.601–1.013)	0.798 (0.615–1.037)		1.715 (0.974–3.018)	1.623 (0.918–2.871)	1.997 (1.128–3.535) *	2.301 (1.286–4.116) **
75–84		0.619 (0.424–0.904) *	0.588 (0.402–0.861) **	0.692 (0.473–1.012)	0.706 (0.483–1.034)		3.865 (2.123–7.035) ***	3.643 (1.99–6.67) ***	4.747 (2.583–8.725) ***	5.374 (2.895–9.975) ***
85≤		0.361 (0.089–1.469)	0.349 (0.086–1.419)	0.396 (0.097–1.610)	0.406 (0.010–1.650)		23.853 (12.038–47.266) ***	23.101 (11.621–45.922) ***	27.770 (13.923–55.392) ***	32.492 (16.133–65.440) ***
Sex										
Male		1.179 (0.441–3.151)	1.204 (0.451–3.217)	1.224 (0.458–3.270)	1.223 (0.458–3.269)		1.094 (0.337–3.551)	1.154 (0.355–3.752)	1.172 (0.362–3.797)	1.134 (0.349–3.683)
Type of insurance (vs. NHI)										
Medical Aid		1.141 (0.854–1.526)	1.124 (0.840–1.502)	1.109 (0.829–1.482)	1.111 (0.831–1.485)		2.206 (1.491–3.264) ***	2.162 (1.460–3.200) **	2.096 (1.415–3.104) **	2.154 (1.455–3.188) ***
CCI										
1–2			1.120 (0.974–1.288)					0.962 (0.7–1.321)		
3¬			1.289 (1.010–1.646) *					1.612 (1.082–2.403) *		
Neoadjuvant chemotherapy										
Yes				1.866 (1.597–2.182) ***					2.521 (1.761–3.608) ***	
Number of sessions										
1–4					1.678 (1.398–2.014) ***					1.534 (0.936–2.512)
5–14					2.575 (1.948–3.404) ***					6.792 (4.140–11.144) ***

Model 1: Univariate Cox regression. ^1^ Model 2: Multivariate Cox regression adjusted for the demographic variables (age group, sex, and type of health insurance). ^2^ Model 3: Multivariate Cox regression adjusted for the demographic variables (age group, sex, and type of health insurance) and CCI. ^3^ Model 4: Multivariate Cox regression adjusted for the demographic variables (age group, sex, and type of health insurance) and neoadjuvant chemotherapy. ^4^ Model 5: Multivariate Cox regression adjusted for the demographic variables (age group, sex, and type of health insurance) and number of NAC sessions. Abbreviations: SLNB, sentinel lymph node biopsy; ALND, axillary lymph node dissection; NHI, National Health Insurance; CCI, Charlson comorbidity index; HR, hazard ratio; CI, confidence interval. * *p*-value < 0.05, ** *p*-value < 0.01, *** *p*-value < 0.001.

**Table 3 healthcare-11-01833-t003:** Association of adjuvant radiotherapy and chemotherapy with lymphedema and death using time-varying regression analysis.

	Adjuvant Radiotherapy				Adjuvant Chemotherapy			
Covariates	Lymphedema ^1^		Death ^2^		Lymphedema ^3^		Death ^4^	
	HR (95% CI)		HR (95% CI)		HR (95% CI)		HR (95% CI)	
Type of surgery (vs. ALND)								
SLNB	0.508 (0.462–0.56) ***	0.513 (0.466–0.565) ***	1.081 (0.868–1.346)	1.081 (0.866–1.350)	0.513 (0.466–0.565) ***	0.514 (0.467–0.566) ***	0.861 (0.691–1.072)	0.828 (0.664–1.031)
Adjuvant radiotherapy								
Yes	1.002 (0.998–1.006)		0.994 (0.987–1.001)					
Number of sessions								
1–10		1.551 (1.356–1.775) ***		1.398 (1.049–1.863) *				
11–20		1.75 (1.389–2.204) ***		0.915 (0.532–1.572)				
21–30		1.424 (1.389–2.204) ***		0.798 (0.555–1.147)				
31–40		1.064 (0.916–1.236)		0.95 (0.689–1.311)				
41–50		-		1.402 (0.502–3.916)				
51 or more		9.574 (4.524–20.258) ***		1.003 (0.408–2.469)				
Adjuvant chemotherapy								
Yes					1.061 (1.053–1.068) ***		0.955 (0.942–0.968) ***	
Number of sessions								
1–5						1.354 (1.199–1.53) ***		0.321 (0.229–0.449) ***
6–10						3.410 (2.987–3.892) ***		0.248 (0.174–0.354) ***
11–15						2.881 (2.268–3.661) ***		0.246 (0.158–0.384) ***
16–20						3.775 (2.806–5.708) ***		0.193 (0.120–0.312) ***
21–25						2.180 (1.574–4.955) **		0.166 (0.093–0.295) ***
26–30						2.638 (1.404–4.955) **		0.411 (0.243–0.696) ***
31 or more						5.016 (1.868–13.468) **		0.257 (0.149–0.444) ***

^1^ Estimate of adjuvant radiotherapy as a time-varying exposure on lymphedema, β = 0.002, *p* = 0.265. ^2^ Estimate of adjuvant radiotherapy as a time-varying exposure on death, β = 0.004, *p* = 0.111. ^3^ Estimate of adjuvant chemotherapy as a time-varying exposure on lymphedema, β = 0.059, *p* < 0.001. ^4^ Estimate of adjuvant chemotherapy as a time-varying exposure on death, β = −0.046, *p* < 0.001. * *p*-value < 0.05, ** *p*-value < 0.01, *** *p*-value < 0.001.

## Data Availability

The datasets analyzed in the current study are not publicly available due to the authorization process by the inquiry committee of research support within the NHIS but are available through the corresponding author on reasonable request after the permission of the NHIS.

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
