# Peer review of "Risk of Lymphedema and Death after Lymph Node Dissection with Neoadjuvant and Adjuvant Treatments in Patients with Breast Cancer: An Eight-Year Nationwide Cohort Study"

_healthcare, 2023, doi:10.3390/healthcare11131833_

Round 1
Reviewer 1 Report (Previous Reviewer 2)
Reviews have been addressed. Accept
Reviews have been addressed. Accept
Author Response
We thank the reviewer for the decision.
Reviewer 2 Report (Previous Reviewer 1)
Thank you for the revised form. Please rephrase the last sentence of your abstract “The findings emphasize the importance of long-term surveillance…to minimize the risk of LE…” because it is a little confusing (in the abstract) how surveillance (without acting?) can minimize the risk of LE and maximise overall survival.
Author Response
We thank the reviewer for the thoughtful feedback. Based on the reviewer's comment, we revised the abstract as follows:
Before:
The findings emphasize the importance of long-term surveillance of patients with breast cancer to minimize the risk of LE and maximize overall survival.
After:
The findings highlight the importance of not only early surveillance before and after surgery, but also long-term surveillance during adjuvant treatment by surgeons and oncologists in order to reduce the risk of LE.
This manuscript is a resubmission of an earlier submission. The following is a list of the peer review reports and author responses from that submission.
Round 1
Reviewer 1 Report
Reference (R) [3], accessed today, is related to data of 2012-2018 with 5-year survival 90.6%. Furthermore, please write “uniformly” (90% and 93% or 90.0% and 93.0%). Besides, R [2] is related to data from Korea and R [3] is related to data from National Cancer Institute. Thus, it is “confusing” to have different data in the same sentence. Lines 31-33 are related to “survival rate”. However, the main conclusion of R [4] is “…did not improve overall survival but reduced the breast cancer recurrence”. Similarly, the main conclusion of R [5] is “…no effect on long-term overall survival, but can improve local control…” and R [6] is not related to increased survival. Furthermore, your sentence “…owing to recent advances in multimodal therapies…” is “partially” true because some benefits could be attributed to breast cancer screening.
Considering that “death” is included in the title, it must be mentioned that female breast cancer (BC) is the fifth cause of cancer death behind lung, colorectal, liver and stomach cancers [1]. The R [2] related to lines 30 and 31 is somewhat “misleading” because it is related to data from Korea and not “worldwide”. Furthermore, according to Number of Cases by Cancer Sites of your country, BC was in the fifth place (behind stomach, colon-rectum, lung and thyroid).
In lines 36 to 37, it is written “The prevalence of LE and its association with treatment modalities show a variance of 20% to > 50% [10-12]”…However, in Rs [12] and [11], related percentages are >7% and >55% respectively. In line 39, it is written “…body mass index and age…”. However, according to R [8], “in multivariable analysis”, obesity was a weaker factor compared to the number of nodes removed and length of neoadjuvant chemotherapy. Similarly, according to R [8], compared to age <50, CIs in women 50-59, 60-69 and ≥70 cross 1. A similar conclusion related to age is mentioned in R [14]. The R [13] concludes that “older age” is “associated with increased odds…” and they are and other references in your paper with the same conclusion.
In lines 40 to 42, you mention “…complete decongestive therapy (CDT) or manual drainage is proven to be effective…”. However, some important references are missing Examples: Marcos AL et al. Lymphedema of the arm after surgery for breast cancer: new physiotherapy. Clin Exp Obstet Gynecol 2012;39:483-8.
https://pubmed.ncbi.nlm.nih.gov/23444749/
Lu SR et al. Role of physiotherapy and patient education in lymphedema control following breast cancer surgery. Ther Clin Risk Manag 2015;11:319-27.
https://pubmed.ncbi.nlm.nih.gov/25750536/
In lines 263-264, it is written “AR did not significantly influence the LE risk, similar to the findings from previous studies [8, 30]”. However, the conclusions of R [30] are not related to this subject. In lines 265-267, it is written “axilla staging during initial surgical resection, which was the standard treatment procedure for almost two decades [33]. However, during the last decade, the surgical trend shifted from ALND to SLNB…” According to UpToDate 2023 (and general knowledge) “The SLNB technique has been developed and validated over the past three decades”.
In lines 283-284, it is written “especially if taxane-based and administered for a long duration [8,36-38], increased the LE risk”. However, R [36] is not a “comparative” study related to chemotherapy duration, R [37] is mainly related to body mass index and R [38] is not related to the duration “per se”.
In lines 287-288, it is written “SLNB is recommended in patients with T1/T2 and multicentric tumors and against T3/T4… [39].”. However, according to UpToDate 2023 “Certain large tumors (eg, T3) may be amenable to SLNB…”.
In lines 302-303, it is written “…studies on the risk factors of LE report that BMI [13,14,40,41] and age [8,22,40] might be associated with LE,…”. However, R [40] has as a main conclusion that “…patients with low breast density appeared to be at a higher risk of developing severe lymphedema”.
In lines 303-304, it is written “In contrast, our study did not find an association between age and LE”. However, similarly, R [8] did not find such an association.
References [42], [43], [44], [46] and [47] are marginally related or not related to the paper and could be removed.
Reviewer 2 Report
Numerous studies and meta-analyses have already shown that the risk of lymphedema is significantly higher with axillary dissection compared with sentinel node biopsy (SLNB). This study, therefore, brings nothing new in this respect. Evaluation of the various concomitant therapies and the risk of lymphedema is certainly interesting, but the subject is not organized and linear. English is extremely convoluted, and the English language of the article needs to be thoroughly revised.
It is questionable whether SLNB can be considered axillary dissection (lines 87-88).
In addition, the analysis of overall survival in a retrospective study evaluating the type of surgery on the axilla appears to be a purely statistical correlation. The indication for axillary surgery cannot disregard the stage of the disease, its aggressiveness, and its response to neoadjuvant chemotherapy, the numerous biases in the retrospective analysis do not allow the results to be considered informative.
It is unclear how surveillance of patients with breast cancer could minimize the risk of lymphedema and maximize overall survival, this observation is not supported by any result of this study (last sentence of the abstract).
Reviewer 3 Report
In this article, the authors used a large sample size and conducted analysis of risk of lymphedema and death after lymph node dissection with neoadjuvant and adjuvant treatment in patients with breast cancer, they compared the survival of SLNB and ALND, and the SLNB is much better that ALND. This is a quite meaning study with comprehensive statistical analysis, all the figures should be changed to clear ones. Minor revisions are required. Here are some questions or suggestions:
1. The figure 1 study design is blurred. Size of words are too small to be recognized.
2. The authors selected patients sample from 2011-2013( a retrospective study), it seems it is quite old data, why not using patients data in recent 3 or 5 years? Or collect more data from 2011-2021?
3. In Table 1, most of the p value is above 0.05, it seems the p value here is not significant, does it affect the result?
4. Figure 3 is too blurred. Please change to clear one.
5. For “ p value” in some places you use capital letter, and in some other places you use low case letter, please use in low case in italic format.
6. It seems the citations are not the latest, if possible, please check the latest related articles from Pubmed.